# Mobile interventions targeting common mental disorders among pregnant and postpartum women: An equity-focused systematic review

Ammar Saad[1,2], Olivia Magwood[2,3], Tim Aubry[4,5], Qasem Alkhateeb[6], Syeda Shanza Hashmi[7], Julie Hakim[8], Leanne Ford[9], Azaad Kassam[10,11,12], Peter Tugwell[1,13,14], Kevin Pottie[1,2,15] *

1 School of Epidemiology and Public Health, University of Ottawa, Ottawa, Ontario, Canada, 2 C.T. Lamont Primary Healthcare Centre, Bruyère Research Institute, Ottawa, Ontario, Canada, 3 Interdisciplinary School of Health Sciences, University of Ottawa, Ottawa, Ontario, Canada, 4 School of Psychology, University of Ottawa, Ottawa, Ontario, Canada, 5 Centre for Research on Educational and Community Services, University of Ottawa, Ottawa, Ontario, Canada, 6 Department of Medicine, Schulich School of Medicine, Western University, London, Ontario, Canada, 7 Department of Psychiatry, University of Toronto, Toronto, Ontario, Canada, 8 Division of Pediatric and Adolescent Gynecology, Department of Obstetrics and Gynecology, Dallas, Texas, United States of America, 9 Rebirth Wellness Centre Inc., London, Ontario, Canada, 10 Department of Psychiatry, University of Ottawa, Ottawa, Ontario, Canada, 11 Pinecrest Queensway Community Health Centre, Ottawa, Ontario, Canada, 12 Ottawa Newcomer Health Centre, Ottawa, Ontario, Canada, 13 Ottawa Hospital Research Institute, Ottawa, Ontario, Canada, 14 Department of Medicine, University of Ottawa, Ottawa, Ontario, Canada, 15 Department of Family Medicine, University of Ottawa, Ottawa, Ontario, Canada

* kpottie@uottawa.ca

## Abstract

### Introduction

Pregnant and postpartum women face major psychological stressors that put them at higher risk of developing common mental disorders, such as depression and anxiety. Yet, their limited access to and uptake of traditional mental health care is inequitable, especially during the COVID-19 pandemic. Mobile interventions emerged as a potential solution to this discontinued healthcare access, but more knowledge is needed about their effectiveness and impact on health equity. This equity-focused systematic review examined the effectiveness and equity impact of mobile interventions targeting common mental disorders among pregnant and postpartum women.

### Methods and results

We systematically searched MEDLINE, EMBASE, PsychINFO and 3 other databases, from date of database inception and until January 2021, for experimental studies on mobile interventions targeting pregnant and postpartum women. We used pooled and narrative synthesis methods to analyze effectiveness and equity data, critically appraised the methodological rigour of included studies using Cochrane tools, and assessed the certainty of evidence using the GRADE approach. Our search identified 6148 records, of which 18

**Funding:** AS received the Bruyère Research Institute Graduate Studentship Award (April 2020). The funders had no role in study design, data collection and analysis, decision to publish, or preparation of the manuscript.

**Competing interests:** The authors have declared that no competing interests exist.

randomized and non-randomized controlled trials were included. Mobile interventions had a clinically important impact on reducing the occurrence of depression (OR = 0.51 [95% CI 0.41 to 0.64]; absolute risk reduction RD: 7.14% [95% CI 4.92 to 9.36]; p<0.001) and preventing its severity perinatally (MD = -3.07; 95% CI -4.68 to -1.46; p<0.001). Mobile cognitive behavioural therapy (CBT) was effective in managing postpartum depression (MD = -6.87; 95% CI -7.92 to -5.82; p<0.001), whereas other support-based interventions had no added benefit. Results on anxiety outcomes and utilization of care were limited. Our equity-focused analyses showed that ethnicity, age, education, and being primiparous were characteristics of influence to the effectiveness of mobile interventions.

## Conclusion

As the COVID-19 pandemic has increased the need for virtual mental health care, mobile interventions show promise in preventing and managing common mental disorders among pregnant and postpartum women. Such interventions carry the potential to address health inequity but more rigorous research that examines patients' intersecting social identities is needed.

## Introduction

Pregnancy is a life experience characterized by major physical and psychological stressors to the childbearing mother [1,2]. If left unaddressed, such stressors could have long-lasting mental health sequelae that affect the mother for years to come and disrupt the entire family ecosystem [3–5]. Common mental disorders (CMD) comprise a range of non-psychotic conditions defined by the presence of two symptom dimensions; depression and anxiety [6,7]. Research shows that depression and anxiety are the most common mental health conditions during pregnancy and postpartum [8,9]. Recent evidence also suggests that the social isolation and uncertainty surrounding the COVID-19 pandemic has increased the occurrence and worsened the severity of depression and anxiety among this population [10–12].

Equity, in a broader sense, is an ethical value of fairness and social justice [13]. As such, health equity is defined as the absence of avoidable health disparities that are judged to be unfair and unjust [14]. The heightened levels of mental health needs among pregnant and postpartum women are often met with major barriers to accessing traditional mental health care [15,16], especially during the COVID-19 pandemic [17,18]. Our preliminary research shows that these barriers are heterogeneous across different subgroups of pregnant and postpartum women and may be linked to their characteristics and social identities [19], magnifying their vulnerability and worsening their outcomes [20].

"*Mobile interventions*" are on the rise globally [21,22], providing a convenient and easily accessible approach to mental healthcare delivery [23]. These interventions utilize different mobile phone features, such as smartphone applications or text messages, to deliver asynchronous care and support without the need for direct communication with healthcare providers [24]. The mechanisms by which mobile interventions elicit change vary depending on their purpose [23]. Examples of such mechanisms include but are not limited to; peer and social support [25]; symptom tracking and monitoring [26]; health promotion and behavioural change [27]; and self-paced psychotherapy and stress management [28]. Evidence suggests that mobile interventions are acceptable for patients [29], and their mental healthcare

providers [30], positioning them as a potential solution to the health inequity among pregnant and postpartum women. Yet more research is needed to understand the effectiveness of these interventions and the impact they carry on the health equity of this population.

## Research objectives

The objective of this equity-focused systematic review is to synthesize quantitative evidence on the effectiveness and equity impact of prevention- and management-based mobile interventions targeting common mental disorders and stress among pregnant and postpartum women.

## Materials and methods

This equity-focused systematic review was registered (PROSPERO ID: CRD42020200828), prepared according to a published protocol [19], and reported using the Preferred Reporting Items for Systematic Reviews (PRISMA) [31] and its equity extension (PRISMA-E) [32] (S1 and S2 Files). We followed a collaborative research approach whereby stakeholders, including pregnant and postpartum women with lived experience of depression and anxiety were engaged throughout different stages of the project [33]. Detailed description of this engagement is presented in S3 File.

### Search strategy and selection criteria

We developed a comprehensive search strategy in consultation with a health sciences librarian (S4 File). We systematically searched Medline, Embase, PsychINFO, and Cochrane CENTRAL (via OVID), CINAHL (via EBSCO), PTSDPubs, as well as citation lists from Web of Science. We searched these databases from inception until June 2020 and updated the search in January 2021. We did not apply any date, language, or setting restrictions to our search. As well, we conducted a focused grey literature search (S4 File), hand-inspected the reference lists of all relevant systematic reviews, and backward-traced all potentially relevant trial registrations and protocols for publications. All records were uploaded to a systematic review management system "Covidence" [34], and two independent reviewers screened all records against our eligibility criteria, using their titles and abstracts, and then full text (Table 1).

### Data analysis

We used a standardized data collection form to extract data from included studies (S5 File). Two reviewers performed this activity, in duplicate and independently. Any discrepancies were resolved by discussion. We critically appraised the methodological rigour of included studies using the revised Cochrane Risk of Bias 2.0 tool (aka. ROB 2.0) for RCTs [36,37], and the Cochrane Risk of Bias in Non-Randomized Studies of Interventions (aka. ROBINS-I) tool for non-randomized studies [38,39]. Visual representations of these assessments were created using the ROBVIS platform [40].

Our statistical analysis plan aimed at calculating a uniform quantitative effect estimate from each included study to facilitate the narrative and pooled synthesis of results. For continuous outcomes (e.g., severity of symptoms), we chose the mean difference (MD) between study arms at follow-up, as this effect estimate measures the comparative effectiveness of the intervention relative to the control group after a period of intervention implementation. We retrieved mean differences from included studies or calculated them using the statistical algorithms presented by Deeks and Higgins on behalf of the Statistical Methods Group of The Cochrane Collaboration [41]. Calculation of mean differences comprised extracting the means, standard deviations, and sample sizes for each study arm from the included study and

**Table 1. Eligibility criteria and outcomes of interest.**

| Study characteristic | Description of the inclusion criteria |
|---|---|
| Population | Pregnant women at any stage of the pregnancy experience (antenatal, perinatal, or postpartum), and regardless of the pregnancy outcome (e.g., full birth, miscarriage, medically induced abortion). We defined the following pregnancy stages: <br>• **Antenatal**: Any time from the onset of pregnancy and until the time point in which delivery is expected [0–36 weeks] <br>• **Perinatal**: The period of time surrounding delivery and until it occurs [36 weeks-delivery] <br>• **Postpartum**: The period of time after delivery and until 12 months after it occurs [delivery-12 months after] |
| Intervention | Mobile interventions targeting common mental disorders and stress: <br>• Interventions that deliver care or support to pregnant women using features supported by mobile technology (e.g., mobile applications, text messaging programs) with the intention of managing or preventing mental health symptoms, psychological distress, or improving access to pregnancy-related or mental health care. <br>• These interventions operate "asynchronously": independent of direct, face-to-face contact with a psychiatrist, psychologist, mental health professional, or primary healthcare provider, and can utilize different mechanisms (e.g., self-management of symptoms, self-management with supported care and peer support, improving cognition and thinking, improving skills and behaviours, providing psychoeducation and therapy or tracking symptoms). |
| Comparison | • Usual or standard care <br>• Controlled intervention <br>• Placebo intervention <br>• No intervention <br>• Waitlisting |
| Outcomes | • Severity of common mental health symptoms <br>• Changes in the occurrence of common mental health disorders <br>• Psychological wellbeing and distress <br>• Utilization of pregnancy related and mental healthcare services |
| Study design | Using the recommendations of the Cochrane Effective Practice and Organization of Care (EPOC) group [35]: <br>• Randomized and quasi-randomized controlled trials (RCTs; qRCTs) <br>• Non-randomized controlled studies (NRS) <br>• Controlled before and after studies (CBA) <br>• Controlled interrupted time series (CITs) and repeated measures studies. |
| Time frame of follow-up | • Short term: [post-intervention to 3 months] <br>• Medium term: [over 3 months to 6 months] <br>• Long term: [over 6 months of follow-up] |

plotting these values into the RevMan 5.4 software. Calculated effect estimates were accompanied by measures of statistical significance (i.e., 95% confidence intervals and p-values). For categorical outcomes (e.g., occurrence of disease), we chose to retrieve a relative risk measure (odds ratio and/ or risk ratio), as well as an absolute risk measure (absolute risk reduction) for each effect estimate. Whenever these values were not reported in included studies, we aimed to calculate them using the same approach we used for continuous outcomes [41]. Similarly, effect estimates of categorical outcomes were accompanied by measures of statistical significance (i.e., 95% confidence intervals and p-values). To ensure reproducibility of our results, we reported the effect estimates that were retrieved, those that were calculated, and the values used to calculate them from each included study.

We assessed clinical homogeneity across studies by standardizing the definitions of their populations, interventions, comparisons, outcomes, and timepoints [42], and comparing these criteria across studies. Whenever possible, we meta-analyzed data by pooling the mean differences at follow-up from clinically-homogeneous studies using a random effects model [43]. We utilized the inverse variance statistical methods embedded in RevMan 5.4 software to meta-analyze data [44]. We assessed statistical heterogeneity using the $I^2$ ($\geq$70%) and Chi

Square tests of independence ($p \leq 0.1$) [45]. Whenever clinical heterogeneity prevented the pooling of results, we synthesized evidence using a narrative approach [46,47]. For our equity-focused analysis, we adapted the PROGRESS+ framework, which stands for place of residence; race, ethnicity, culture, and language; occupation; gender and sex; religion; education; socio-economic status; social capital; as well as personal characteristics associated with discrimination [48]. We defined "equity evidence" as any effect estimate that can be linked to a PROGRESS+ characteristic, and "equity impact" as any gradient in effect estimates when adjusting for a PROGRESS+ characteristic. Our primary equity analysis focused on ethnicity and race; age; socioeconomic status; social capital; and experience of intimate partner violence [19]. Exploratory equity evidence was reported for other available PROGRESS+ characteristics. Statistical significance was set at the $p < 0.05$ threshold, whereas clinical importance was assessed by reviewing the literature on minimal clinically important differences (MCID) of the tools measuring the outcomes, and by engaging providers and patients in the decision-making process [49]. Finally, we used the Grading of Recommendations, Assessment, Development and Evaluation (GRADE) methodology to assess the certainty of our evidence [50].

## Results

### Search results and characteristics of included studies

Our search yielded a total of 6476 records, of which 4210 records were screened independently after the removal of duplicates (Fig 1). The inter-rater reliability was assessed with a random sample of n = 100 records and Cohen's kappa coefficient of 0.9 proved high screening reliability. We screened 48 studies using their full-text publications and included n = 18 in our quantitative analysis, representing a cumulative sample size of N = 7,181 pregnant or postpartum women. A total of n = 14 studies linked their results to one or more PROGRESS+ characteristic and were included in our equity-focused analysis.

The characteristics of included studies are presented in Table 2. In summary, thirteen included studies followed a randomized controlled study design, four followed a non-randomized controlled study design, and one followed a quasi-randomized controlled study design. The geographical location of the studies was diverse, and two studies required translation. Ten studies examined the effectiveness of prevention-based mobile interventions [51–60], and eight studies examined management-based interventions [61–68]. Outcome domains, measurement tools, and timing of follow-up are presented in Table 2, along with the quantitative statistical analysis method used in each included study, and effect estimate values we calculated.

The majority of included studies were judged to have high risk of bias (Figs 2 and 3). Visual representations of critical appraisal assessments categorized by outcome are presented in S6 File. As well, GRADE Evidence Profiles for each outcome are presented in S7 File and provided alongside effectiveness findings in text. In summary, effectiveness results of mobile interventions targeting the prevention of mental health disorders were judged to be of higher certainty relative to interventions that target the management of mental health disorders. Certainty assessments ranged from high to very low and depended on the outcome and comparison (S7 File).

### The effectiveness of prevention-based mobile interventions

Results on prevention-based mobile interventions showed their potential to prevent depression and psychological stress. Clinical heterogeneity between studies prevented pooling results except for one instance of two studies (Fig 4). Heterogeneity mainly arose due to variability in outcome measurement tools and the pregnancy stage in which the interventions were

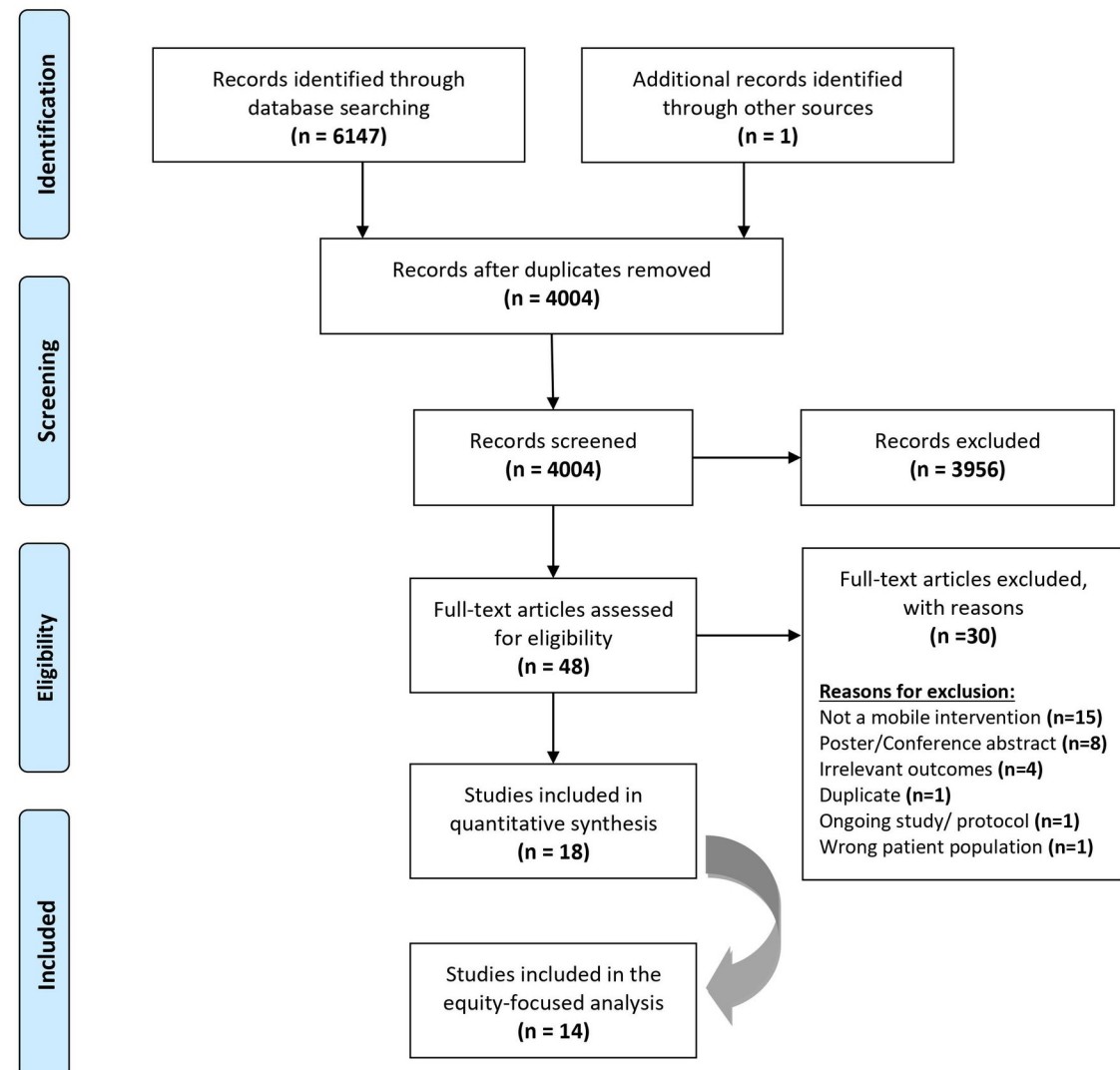

**Fig 1. PRISMA flow diagram of study screening and selection.**

delivered. Results from studies examining prevention-based mobile interventions are presented in Table 3. In the **antenatal** stages of pregnancy, one study measured changes in the occurrence of depression and reported a statistically significant and clinically important

**Table 2. Characteristics of included studies.**

| Study ID | Study design | Setting | Participants | Intervention | Comparison |
|---|---|---|---|---|---|
| Chan et al. 2019 [51] | Randomized controlled trial | Kwong Wah Hospital (KWH) Hong Kong, China | First-time pregnant Chinese women Pregnancy stage: Antenatal | iParent app: Smartphone application providing pregnancy-related health promotion. The app provided a platform to allow asking pregnancy-related questions. Purpose: Preventing postnatal depression. | Standard antenatal care including a 4-session nurse-led antenatal course |
| Cheng et al. 2016 [52] | Randomized controlled trial | Hospital affiliated obstetric clinic Kaohsiung City, Taiwan | Expecting Taiwanese pregnant women Pregnancy stage: Perinatal | Smartphone application providing women with the means to connect and talk with peers (postpartum women). Purpose: Preventing postpartum depression and stress. | Standard perinatal care |
| Chyzzy 2019 [53] | Randomized controlled trial | Community-based young parents' agencies Toronto, Canada | Expecting pregnant adolescents 16–24 years Pregnancy stage: Perinatal | Smartphone application providing women with the means to talk (via voice and messages) with peer mentors (postpartum women) Purpose: Preventing postpartum depression | Standard community support and care services |
| Gong et al. 2020 [54] | Non-randomized study design: Non-randomized controlled trial | Three public hospitals. Jiangmen City, Guangdong Province, China | Chinese pregnant women Pregnancy stage: Antenatal | Text messaging program consists of one-way messages promoting health and behavioural changes, as well as reminding participants of their routine appointments Purpose: Preventing depression during pregnancy | Standard antenatal care |
| Jareethum et al. 2008 [55] | Randomized controlled trial | The antenatal care unit of the Siriraj Hospital Bangkok, Thailand | Thai Pregnant women planning to deliver at the hospital Pregnancy stage: Antenatal | Text messaging program providing antenatal support and education about pregnancy symptoms Purpose: Increasing satisfaction with care and preventing anxiety | Standard antenatal care from the hospital |
| Lee and Kim 2017 [56] | Non-randomized study design: Non-randomized controlled trial | Four maternal hospitals Daegu, South Korea | First-time South Korean Postpartum mothers Pregnancy stage: Postnatal | Smartphone application with an interactive platform and newsfeed feature providing postpartum women with postpartum-related health management education and emergency contacts. Purpose: Preventing postpartum depression and increasing confidence | Standard postpartum discharge care from the hospital |
| Mauriello et al. 2016 [57] | Randomized controlled trial | Three federally funded health centre organizations Connecticut, Rhode Island, and New York, United States | English and Spanish speaking pregnant women Pregnancy stage: Antenatal | Healthy Pregnancy Step By Step: iPad-delivered application providing women with risk stage-matched and tailored guidance on theoretically and empirically determined behavioural change strategies. Purpose: Preventing postpartum stress and promoting stress management | Standard antenatal care from community organizations and behavioural change brochures |
| Shorey et al. 2019 [58] | Randomized controlled trial | Postnatal ward of a local tertiary hospital National University Hospital region, Singapore | Postpartum mothers at risk of depression Pregnancy stage: Postnatal | Mobile-based program providing mothers with the means to communicate with a trained peer volunteer using text-messages, phone calls, or applications depending on the mother's preference Purpose: Preventing postnatal depression and buffering the negative effects of childbirth | Standard postnatal care from the hospital |

*(Continued)*

**Table 2.** (Continued)

| Study ID | Study design | Setting | Participants | Intervention | Comparison |
|---|---|---|---|---|---|
| Shorey et al. 2017 [59] | Randomized controlled trial | Maternity ward of a local tertiary hospital National University Hospital region, Singapore | Postpartum mothers and their partners Pregnancy stage: Postnatal | Home But Not Alone: Mobile application providing mothers and their partners with psychoeducation and health promotion and reminding them of their appointments Purpose: Decrease the risk of postnatal depression and improve parenting self-efficacy | Standard postnatal care provided by the hospital |
| Tsai et al. 2018 [60] | Non-randomized study design: Comparative cohort study | Obstetrics outpatient clinic at a medical center Tainan, Taiwan | Taiwanese pregnant women with low-risk pregnancy Pregnancy stage: Antenatal | Smartphone application where pregnant women can upload and access their antenatal care records. The application provided women with health promotion and journals for the self-management of symptoms. Access was available through the internet as well as the smartphone. Purpose: Improving self-efficacy and prevent psychological stress | Standard antenatal support and education |
| Baumel et al. 2018 [61] | Non-randomized study design: Historically controlled study | The Adult Outpatient Department in the Zucker Hillside Hospital New York, United States | New mothers with postpartum depression Pregnancy stage: Postnatal | 7Cups: Smartphone application providing women with the means to contact community peers or "past survivors", a personalized progress map, and psychotherapy, such as gratitude exercises, mindfulness, psychoeducation, exercises drawn from principles of acceptance and commitment therapy. Purpose: Manage postpartum depression and support women with mood disorders | Standard care or "treatment as usual" |
| Carissoli et al. 2017 [62] | Randomized controlled trial | At antenatal childbirth classes organized at the obstetrics wards of Saronno, Gallarate and Busto Arsizio Hospitals Province of Varese, Italy | First-time Italian pregnant women approaching birth Pregnancy stage: Perinatal | BenEssere Mama: Smartphone application providing mothers with daily relaxation and guided imagery exercises alongside mood journaling. Purpose: Help mothers manage their affective state and improve their psychological wellbeing | Standard perinatal care |
| Constant et al. 2014 [63] | Randomized controlled trial | Two non-governmental organizations (NGOs) and two public sector primary care clinics Cape Town, South Africa | Pregnant women undergoing medical abortion Pregnancy stage: Post-abortion | Uniform text messaging program with health information about managing symptoms and side effects, alongside routine abortion care. Purpose: Managing anxiety and emotional discomfort | Routine abortion care (including the provision of 200-mg mifepristone on site and self-administration of 800-mcg misoprostol at home) was provided to women in both study arms |
| Dennis-Tiwary et al. 2017 [64] | Randomized controlled trial | Large urban hospital New York, United States | Pregnant women Pregnancy stage: Antenatal | Personal Zen: smartphone application that utilizes an attention bias modification training (ABMT) protocol with video game-like features such as animated characters and sound effects. Purpose: Manage anxiety and stress | Controlled application providing placebo training (PT) |

*(Continued)*

**Table 2.** (*Continued*)

| Study ID | Study design | Setting | Participants | Intervention | Comparison |
|---|---|---|---|---|---|
| Hantsoo et al. 2018 [65] | Randomized controlled trial | Urban ambulatory prenatal clinic within an academic medical center Pennsylvania, United States | Pregnant women from racial-ethnic minority groups with low incomes who were experiencing depressive symptoms. Pregnancy stage: Antenatal | GINGER.IO Mood tracking and alert app: Smartphone application that monitors participants' mood through daily surveys and physical activity trends and alerts providers of care of worsened mood Purpose: Enhancing management of mood symptoms by improving mental health care delivery | Participants in both study arms received standard of care that includes a controlled smartphone application that allows access to the patient portal (PP) |
| Jannati et al. 2020 [66] | Randomized controlled trial | Three health care centers affiliated with Kerman University of Medical Sciences Kerman, Iran | New mothers with postpartum depression Pregnancy stage: postnatal | Happy Mom: Smartphone application providing women with cognitive behavioural therapy in the form of 8 lessons that read like a story Purpose: Managing postpartum depression symptoms | Standard postpartum care as needed by mothers |
| Prasad 2018 [67] | Quasi-randomized controlled trial Quasi-random allocation scheme: days of the week | Local physician clinics Texas, United States | New mothers with postpartum depression Pregnancy stage: postnatal | VeedaMom: Smartphone application providing mothers with mindfulness and meditation exercises, videos that provide psychoeducation based on acceptance and commitment therapy (ACT) and dialectical behavioural therapy (DBT), as well as social support, in-app journaling, and mood tracking. Purpose: Managing symptoms of depression after delivery and improving psychological wellbeing | Standard postnatal care in the form of booklets and resources for postnatal mothers provided by the state |
| Sawyer et al. 2019 [68] | Randomized controlled trial | Child and Family Health Service (CaFHS) community clinics Adelaide and South, Australia | New mothers with depression and parenting problems Pregnancy stage: Postnatal | eMums Plus: Smartphone application that provides mothers with means to chat with mothers and post questions about their experience. A time-sensitive guidance and health reminders, as well as resources and contacts. All chats were monitored by a nurse Purpose: Managing depression symptoms and improving parenting skills | Standard postnatal care that includes an in-home visit by the nurse |

| Study ID | Sample size [a] | Outcomes | Outcome measurement tool | Outcome measurement time [b] | Quantitative analysis method used Study results used in our analysis [calculated results] [c] |
|---|---|---|---|---|---|
| **Mobile interventions targeting the *prevention* of mental health disorders** | | | | | |
| Chan et al. 2019 [51] | 660 IG: 330 CG: 330 | Severity of depression symptoms | The validated Chinese version of the 10-item Edinburgh Postnatal Depression Scale (EPDS) | Short term | Analysis of covariance (ANCOVA) MD at follow-up; 95% CI; p-value |
| | | Severity of anxiety symptoms | The anxiety subscale of the Depression Anxiety Stress Scale (DASS) | | |
| | | Severity of psychological stress | The stress subscale of the Depression Anxiety Stress Scale (DASS) | | |

(*Continued*)

**Table 2.** (*Continued*)

| Study ID | Study design | Setting | Participants | Intervention | Comparison |
|---|---|---|---|---|---|
| Cheng et al. 2016 [52] | 140 IG: 70 CG: 70 | Severity of depression symptoms | The Edinburgh Postnatal Depression Scale—Chinese version | Short term | Analysis of covariance (ANCOVA) MD at follow-up; SDs; p-value |
| | | Severity of psychological stress | The Perceived Stress Scale—Chinese version | | |
| Chyzzy 2019 [53] | 40 IG: 21 CG: 19 | Severity of depression symptoms | The Edinburgh Postnatal Depression Scale (EPDS) | Short term | Independent two-sample t test M and SD for each study arm; p-value [MD at follow-up; 95% CI; p-value] |
| | | Severity of anxiety symptoms | The State-Trait Anxiety Inventory (STAI) | | |
| | | Pregnancy related or mental health service utilization | Number of healthcare visits using a questionnaire | | |
| Gong et al. 2020 [54] | 4501 IG: 1739 CG: 2762 | Occurrence of depression | A cut off = 9 on the Edinburgh Postnatal Depression Scale (EPDS)—Chinese version | Short term | Chi square test; univariate logistic regression N and % of participants with depression at each study arm; OR of not having depression [OR of having depression; RR; ARR; 95% CI; p-value] |
| Jareethum et al. 2008 [55] | 68 IG: 34 CG: 34 | Severity of anxiety symptoms | Questionnaire [unspecified] | Short term | Student t-test M and SD for each study arm; p-value [MD at follow-up; 95% CI; p-value] |
| Lee and Kim 2017 [56] | 81 IG: 40 CG: 41 | Severity of depression symptoms | The Korean version of the Edinburgh Postnatal Depression Scale | Short term | Paired t-test M and SD for each study arm; p-value [MD at follow-up; 95% CI; p-value] |
| Mauriello et al. 2016 [57] | 335 IG: 169 CG: 166 | Pregnancy related or mental health service utilization | Time spent practicing stress management using a structured questionnaire | Short term, medium term | Generalized estimating equations (GEEs) M and SD for each study arm; p-value [MD at follow-up; 95% CI; p-value] |
| Shorey et al. 2019 [58] | 138 IG: 69 CG: 69 | Severity of depression symptoms | The Edinburgh Postnatal Depression Scale (EPDS) | Short term | Repeated measures analysis using a linear mixed model MD at follow-up; 95% CI; p-value |
| | | Severity of anxiety symptoms | The State-Trait Anxiety Inventory (STAI) | | |
| Shorey et al. 2017 [59] | 125 IG: 63 CG: 62 | Severity of depression symptoms | The Edinburgh Postnatal Depression Scale (EPDS) | Short term | Linear mixed-effect model analyses MD at follow-up; 95% CI; p value |
| Tsai et al. 2018 [60] | 155 IG: 80 CG: 75 | Severity of psychological stress | The Chinese Pregnancy Stress Rating Scale-36 (PSRS-36) | Short term | Analyses of covariance (ANCOVA) M and SD for each study arm; p-value [MD at follow-up; 95% CI; p-value] |
| Mobile interventions targeting the *management* of mental health disorders | | | | | |
| Baumel et al. 2018 [61] | 36 IG: 19 CG: 17 | Severity of depression symptoms | The Edinburgh Postnatal Depression Scale (EPDS) | Short term | Paired t-tests M and SD for each study arm [MD at follow-up; 95% CI; p-value] |

(*Continued*)

**Table 2.** (Continued)

| Study ID | Study design | Setting | Participants | Intervention | Comparison |
|---|---|---|---|---|---|
| Carissoli et al. 2017 [62] | 78 IG: 35 CG: 43 | Psychological wellbeing | The Psychological Wellbeing Questionnaire (PWB)—Italian version | Short term | Repeated measures analysis of variance (ANOVA) M and SD for each study arm; p-value [MD at follow-up; 95% CI; p-value] |
| Constant et al. 2014 [63] | 469 IG: 234 CG: 235 | Severity of anxiety symptoms | The Hospital Anxiety and Depression Scale (HADS) | Short term | Linear regression models M and SD for anxiety measures; Beta coefficients for stress measures [MD at follow-up; 95% CI; p-value for anxiety measures; Beta coefficients were reported as is for stress measures] |
| | | Severity of psychological stress | The Impact of Event Scale (IES) | | |
| Dennis-Tiwary et al. 2017 [64] | 29 IG: 15 CG: 14 | Severity of psychological stress | Lab-acquired cortisol saliva sample (ug/dl) | Short term | Analysis of covariance (ANCOVA) ANCOVA F-test; p-value [Results reported as is] |
| Hantsoo et al. 2018 [65] | 72 IG: 48 CG: 24 | Pregnancy related or mental health service utilization | Number of provider phone calls that addressed mental health using electronic health record review | Short term | Analysis of covariance (ANCOVA) M and SD for each study arm [MD at follow-up; 95% CI; p-value] |
| Jannati et al. 2020 [66] | 78 IG: 39 CG: 39 | Severity of depression symptoms | The Edinburgh Postnatal Depression Scale (EPDS)—Persian version | Short term | Paired t-test; linear regression analysis M and SD for each study arm [MD at follow-up; 95% CI; p-value] |
| Prasad 2018 [67] | 43 IG: 23 CG: 20 | Severity of depression symptoms | The Edinburgh Postnatal Depression Scale (EPDS) | Short term | Mixed analysis of variance (ANOVA) M and SD for each study arm [MD at follow-up; 95% CI; p-value] |
| | | Psychological wellbeing | The World Health Organization Quality of Life-BREF (WHOQOL-BREF) tool | | |
| Sawyer et al. 2019 [68] | 133 IG: 72 CG: 61 | Severity of depression symptoms | The Edinburgh Postnatal Depression Scale (EPDS) | Short term | Linear generalized estimating equations (GEEs) M and SD for each study arm [MD at follow-up; 95% CI; p-value] |
| | | Pregnancy related or mental health service utilization | Percentage of women who visited the emergency department 2 or more times in the past 6 months using a standardized questionnaire | | |

[a] IG: Intervention group; CG: Control group.

[b] Short term: post-intervention to 3 months (inclusive); medium term: > 3 month to 6 months (inclusive); long-term: > 6 months.

[c] M: Mean; SD: Standard deviation; MD: mean difference; N: Number of participants with outcome; OR: Odds ratio; RR: Risk ratio; ARR: Absolute risk reduction; 95% CI: 95% confidence interval.

decrease in the odds of being screened positive for depression among the intervention group compared to the usual care group in the short term (OR = 0.51 [95% CI 0.41 to 0.64]; RR = 0.56 [95% CI 0.46 to 0.68]; absolute risk reduction RD: 7.14% [95% CI 4.92 to 9.36]; p<0.001; GRADE certainty: low) [54]. Severity of depression was measured in one study that found statistically significant, but not clinically important, improvement at 4-weeks follow-up (GRADE certainty: high) [51]. As well, evidence on anxiety from two studies was limited and did not reach statistical significance nor clinical importance [51,55], whereas evidence on psychological stress was mixed but showed potential; one study among antenatal women showed

**Fig 2. Risk of bias assessment in randomized controlled trials of effectiveness (ROB 2.0).**

a statistically significant and clinically important improvements in stress levels compared to usual care at 12 weeks follow-up (MD = -11.12 [95% CI -17.19 to -5.05]; p<0.001; GRADE certainty: very low) [60], whereas another study found no added benefit at 4 weeks of follow-up [51].

In the **perinatal** stages of pregnancy, a meta-analysis of two studies providing peer support mobile applications (Fig 4) showed short term improvements in depression severity that were statistically significant and clinically important (MD = -3.07 [95% CI -4.68 to -1.46]; p<0.001;

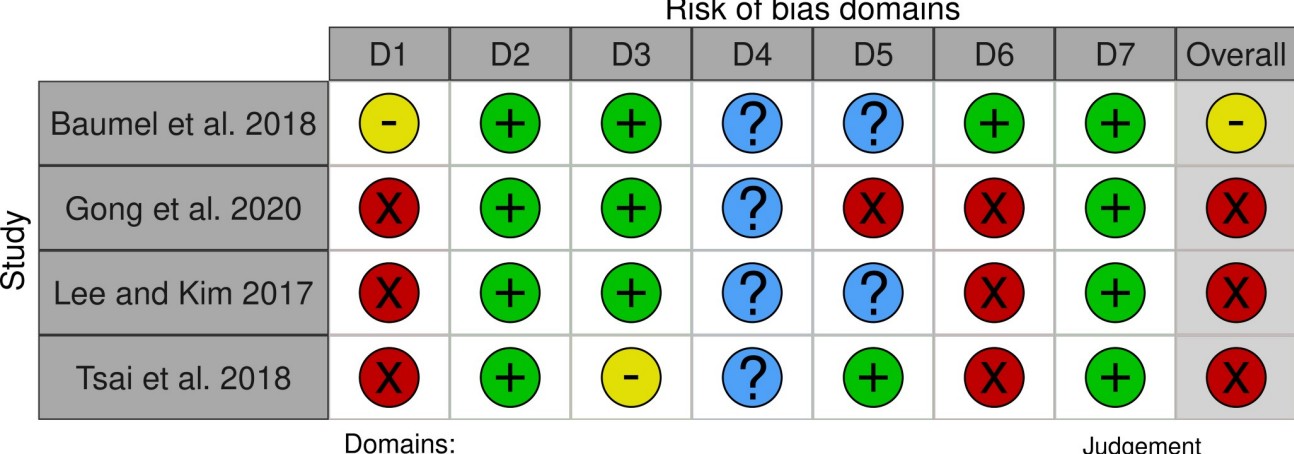

**Fig 3. Risk of bias in non-randomized controlled studies of effectiveness (ROBINS-I).**

GRADE certainty: very low) [52,53]. Pooled results were judged to be statistically homogeneous ($I^2$ = 30%; $Chi^2$ p = 0.23). Although one study showed statistically significant improvements in psychological stress associated with a mobile intervention compared to usual care [52], this improvement was not clinically important (Table 3). Further, the impact of mobile interventions on anxiety and utilization of care [53] from one study was not statistically significant nor clinically important.

Results of mobile interventions delivered to **postpartum** women showed their limited effectiveness on depression and anxiety symptoms. Three studies showed non-clinically important improvements in depression severity [56,58,59], albeit two were statistically significant [56,58]. Clinical heterogeneity prevented pooling of results and GRADE evidence certainty ranged from moderate to very low. Similarly, one study showed improvements in anxiety symptoms that were not statistically significant nor clinically important [58].

### The effectiveness of management-based mobile interventions

Results on management-based mobile interventions showed their potential only among postpartum women. Clinical heterogeneity, arising from variability in intervention design and

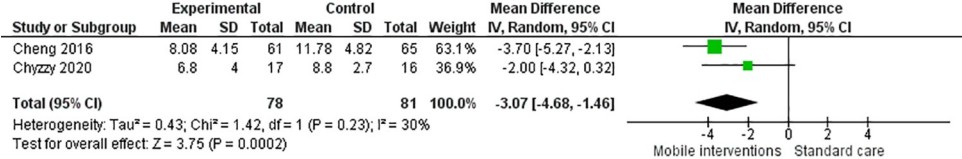

**Fig 4. Forest plot of comparison: peer support mobile applications vs standard of care, outcome: severity of depression symptoms measured using the Edinburgh Postnatal Depression Scale (EPDS)—short term.** IV, Random: Inverse Variance methods, random effects model; CI: confidence interval.

**Table 3. Effectiveness results of prevention-based mobile interventions.**

| Outcome and outcome domain | Study ID | Effect estimate [95% CI or p value] | Statistical significance | Clinical importance | GRADE certainty |
|---|---|---|---|---|---|
| **Prevention-based mobile interventions** | | | | | |
| **Antenatal interventions** | | | | | |
| Depression severity | Chan 2019 [51] | MD = -0.65 [-1.29 to 0.0] | Yes | No | High |
| Depression occurrence | Gong 2020 [54] | OR = 0.51 [0.41 to 0.64] | Yes | Yes | Low |
| Anxiety severity | Chan 2019 [51] | MD = 0.01 [-0.30 to 0.32] | No | No | Moderate |
| Anxiety severity | Jareethum 2008 [55] | MD = -1.01 [-2.28 to 0.26] | No | No | Very low |
| Psychological stress | Chan 2019 [51] | MD = 0.07 [-0.35 to 0.50] | No | No | Moderate |
| | Tsai 2018 [60] | MD = —11.12 [-17.19 to -5.05] | Yes | Yes | Very low |
| Utilization of care | Mauriello 2016 [57] | MD = 15.77 [0.12 to 31.42] | No | No | Very low |
| **Perinatal interventions** | | | | | |
| Depression severity | Cheng 2016 [52] Chyzzy 2020 [53] | MD = -3.07 [-4.68 to -1.46] | Yes | Yes | Very low |
| Anxiety severity | Chyzzy et al. 2020 [53] | MD = -2.0 [-7.71 to 3.71] | No | No | Low |
| Psychological stress | Cheng et al. 2016 [52] | MD = -3.52 [-4.95 to -2.09] | Yes | No | Very low |
| Utilization of care | Chyzzy et al. 2020 [53] | MD = 5.3 [-6.97 to 17.57] | No | No | Low |
| **Postpartum interventions** | | | | | |
| Depression severity | Shorey 2019 [58] | MD = -2.11 [-4.0 to -0.3] | Yes | No | Moderate |
| Depression severity | Shorey 2017 [59] | MD = -0.69 [-1.66 to 0.29] | No | No | Moderate |
| Depression severity | Lee 2017 [56] | MD = -2.68 [-4.86 to -0.5] | Yes | No | Very low |
| Anxiety severity | Shorey 2019 [58] | MD = -2.45 [-9.9 to 5.0] | No | No | Moderate |

pregnancy stage in which the inteventions were delivered prevented pooling results, except for one instance of two studies on support-based mobile interventions (Fig 5). Results of studies examining management-based mobile interventions are presented in Table 4. Evidence on the impact of mobile interventions delivered **antenatally** or **perinatally** was limited and showed no added benefits on psychological stress [62,64], or seeking mental healthcare [65]. GRADE certainty of antenatal and perinatal evidence ranged from low to very low.

Evidence on **postpartum** mobile interventions was mixed; a meta-analysis of two support-based mobile applications (Fig 5) showed no statistically significant nor clinically important improvements in the severity of depression symptoms compared to standard care in the short term (MD = -0.93 [95% CI -2.08 to 0.21]; p = 0.11; GRADE certainty: very low) [67,68]. Pooled results were judged to be statistically homogeneous ($I^2$ = 0%; $Chi^2$ p = 0.92). This result was supported by another non-randomized study of a similar intervention that showed trivial benefits compared to usual care [61]. However, one study of a cognitive behavioural therapy (CBT) mobile application among postpartum women with depression showed statistically significant and clinically important improvement in the severity of their symptoms at 2-month follow-up (MD = -6.87 [95% CI -7.92 to -5.82]; p<0.001; GRADE certainty: very low) [66].

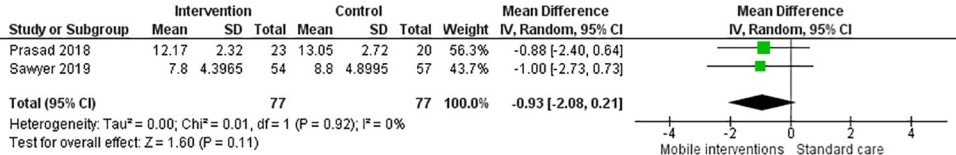

**Fig 5. Forest plot of comparison: support-based psychotherapy applications vs standard of care, outcome: severity of depression symptoms measured using the Edinburgh Postnatal Depression Scale (EPDS)—short term.** IV, Random: Inverse Variance methods, random effects model; CI: confidence interval.

**Table 4. Effectiveness results of management-based mobile interventions.**

| Outcome and outcome domain | Study ID | Effect estimate [95% CI or p value] | Statistical significance | Clinical importance | GRADE certainty |
|---|---|---|---|---|---|
| **Management-based mobile interventions** | | | | | |
| **Antenatal interventions** | | | | | |
| Biological stress | Dennis-Tiwary 2017 [64] | F 1,22 = 4.96 [p = 0.037] | Yes | No | Low |
| Utilization of care | Hantsoo 2018 [65] | MD = 0.88 [-1.08 to 2.84] | No | No | Very low |
| **Perinatal interventions** | | | | | |
| Psychological wellbeing | Carissoli 2017 [62] | Autonomy F = 5.725 [p<0.05] | Yes | No | Very low |
| **Postpartum interventions** | | | | | |
| Depression severity | Prasad 2018 [67] Sawyer 2019 [68] | MD = -0.93 [-2.08 to 0.21] | No | No | Very low |
| Depression severity | Baumel 2018 [61] | MD = 2.82 [-0.49 to 6.13] | No | No | Low |
| Depression severity | Jannati 2020 [66] | MD = -6.87 [-7.92 to -5.82] | Yes | Yes | Very low |
| Anxiety severity | Constant 2014 [63] | MD = -1.30 [-2.33 to -0.27] | Yes | No | Very low |
| Psychological wellbeing | Prasad 2018 [67] | MD = 0.23 [-2.73 to 0.73] | Yes | No | Very low |
| Psychological stress | Constant 2014 [63] | Avoidance B = -1.8 [-3.2 to -0.4] Intrusion B = -1.4 [-2.9 to 0.2] | Yes No | No No | Very low |
| Utilization of care | Sawyer 2019 [68] | [intervention vs control; p value] GP visits: 78% vs 63% [p = 0.09] A&E visits: 15% vs 4% [p = 0.04] Online care: 67% vs 46% [p = 0.03] | No Yes Yes | No No No | Very low |

Evidence on postpartum anxiety, psychological stress, and utilization of care showed a trend of improvements that were statistically significant but not clinically important; among women who underwent medically-induced abortion, one study of a text messaging intervention found statistically significant but not clinically important decrease in levels of anxiety relative to usual discharge care at 2–3 weeks after abortion (MD = -1.30 [95% CI -2.33 to -0.27]; p = 0.01; GRADE certainty: very low) [63]. When measuring subjective stress in the same study investigators found that women receiving the intervention reported lower levels of avoidance-based stress but not intrusive-based stress after adjusting for baseline anxiety (β = -1.8 [95% CI -3.2 to -0.4]; p = 0.015; GRADE certainty: very low) [63]. Another study of a mobile application showed statistically significant improvements on psychological stress that, similarly, did not reach clinical importance levels (GRADE certainty: very low) [67]. Finally, one study of a peer support mobile application found that postpartum women receiving the intervention were more likely to visit their general practitioner, visit the A&E department, and seek pregnancy-related online resources when needed. These findings were not clinically important, and lack of data prevented synthesizing results [68].

## The health equity impact of mobile interventions

Table 5 represents a heatmap of our equity evidence. Overall, women's ethnicity was the most examined characteristic, followed by age, and being primiparous (first-time mother).

Details about our equity results are presented in a compartmentalized (Outcome x PROGRESS+ characteristic) table (S8 File). In summary, evidence showed that mobile interventions elicited significant improvements in mental health outcomes across East and South East Asian ethnicities, such as Chinese [51,54], Taiwanese [52,60], South Korean [56], and Thai [55], as well as West Asian ethnicities, such as Persian [66]. These improvements were variable across ethnicities, but clinical heterogeneity prevented subgroup analyses. Furthermore, exploring evidence that is linked to age, education, and being primiparous (first-time mothers), showed that these characteristics had mixed associations to the degree of statistical significance and

**Table 5. Heatmap of equity evidence.**

| Equity findings | Severity of symptoms | Psychological wellbeing and distress | Occurrence of psychiatric illnesses | Utilization of pregnancy and psychiatric care |
|---|---|---|---|---|
| Place of residence | 1 | 0 | 0 | 0 |
| Race, ethnicity, culture | 9 | 9 | 1 | 3 |
| Occupation | 1 | 0 | 1 | 0 |
| Gender, sex | 0 | 0 | 0 | 0 |
| Religion | 0 | 0 | 0 | 0 |
| Education | 4 | 0 | 1 | 0 |
| Socioeconomic status | 1 | 0 | 1 | 3 |
| Social capital | 3 | 0 | 1 | 0 |
| + Age | 8 | 0 | 1 | 5 |
| + Disability | 0 | 0 | 0 | 0 |
| + Time-dependent: Primiparous | 3 | 7 | 0 | 0 |
| + Time-dependent: IPV | 0 | 0 | 0 | 0 |
| + Discrimination | 1 | 0 | 0 | 0 |

Hues of the colour red increased in darkness with an increase in the number of equity evidence.

clinical importance of the effectiveness of mobile interventions (S8 File). Finally, evidence on the association of socioeconomic status, social capital, and experience of intimate partner violence to the effectiveness of mobile interventions was limited.

## Discussion

COVID-19 and the public health restrictions that ensued have shifted the approach by which mental health care is accessed and delivered. Exploring novel approaches to mental healthcare delivery requires a comprehensive understanding of their risks and benefits, as well as their equity impact among different populations. Our equity-focused systematic review aimed to provide knowledge users with an equity-focused evidence base on the impact of mobile interventions targeting the prevention and management of common mental disorders among pregnant and postpartum women.

Our findings highlight the clinical impact of prevention-based mobile interventions on lowering the severity and occurrence of depression throughout pregnancy [51–54], as well as the potential they carried postpartum [56,58,59]. While findings on preventing psychological stress showed promise [52,60], evidence on preventing anxiety and promoting utilization of care was inconclusive. Furthermore, findings on management-based mobile interventions were limited during pregnancy and only showed promise postpartum; while a smartphone application delivering cognitive behavioral therapy (CBT) was clinically effective in managing the severity of postpartum depression [66], interventions utilizing other mechanisms, such as peer support and psychoeducation, did not show added benefit [61,67,68]. These findings highlight two characteristics of mobile interventions that require further investigation; timing of intervention delivery and the mechanisms under which the intervention operates. Scientific realism is well-positioned to explore these characteristics and shed light on the actors of change in the context of implementing mobile interventions [69].

Our equity evidence highlights ethnicity as a characteristic of influence among pregnant and postpartum women using mobile interventions. Findings suggest that these interventions carry the potential to elicit change across East and West Asian ethnicities [51,52,54–56,60,66]. However, study heterogeneity prevented synthesizing results and conducting credible

subgroup analyses. Of note, we recognize that ethnicity is a social identity that comprises multiple intersecting constructs of a shared culture [70], such as values and principles, practices and social norms, and religion and culture [71]. Understanding the influence of ethnicity, therefore, requires applying an intersectionality lens to our analysis to gain a deeper understanding of these constructs and how they mutually intersect and influence social disadvantage [72]. Moreover, our findings highlight other PROGRESS+ characteristics that require further investigation, such as age, education, and being primiparous (first time mothers).

This review is unique in that it used a comprehensive search strategy to capture records from different information sources, such as electronic bibliographic databases (i.e., Medline, Embase, PsycINFO, Cochrane CENTRAL, PTSDPubs, and Web of Science), trial registries, study protocols, reference lists of relevant studies and systematic reviews, as well as the grey literature. We used rigorous methodology to screen search records, extract data from included studies, critically appraise each of our results, and assess the certainty of our evidence. To the best of our knowledge, this review is the first to apply an equity lens and explore the health equity impact of mobile interventions among pregnant and postpartum women. Our findings were reported transparently and presented using innovative data presentation techniques (i.e., heat maps and compartmentalized tables), to facilitate future dissemination to and interpretation by knowledge users, such as providers of care, policy makers, and patients. Furthermore, the robustness of our methods and clarity in our reporting positions this review to serve as the starting point for future work that can replicate our methods and build upon our findings to advance knowledge and use of mobile interventions among pregnant and postpartum women. Finally, we engaged pregnant and postpartum women with lived experience of common mental disorders in interpreting our evidence, deciding on clinically important findings, and developing our knowledge translation strategies. This work, however, is not without limitations; Firstly, while our search strategy was iteratively developed in consultation with a health sciences librarian and experts in the field of knowledge synthesis, we did not use a structured process to peer review its components in duplicate [73]. Secondly, restricting our eligibility criteria to controlled studies have allowed us to synthesize results on the comparative effectiveness of mobile interventions relative to controlled conditions, but have also precluded the inclusion of longitudinal single-arm studies, which may have enriched our findings. Expanding our inclusion criteria to curate evidence from other study designs is, therefore, needed in future updates of this review. Thirdly, while our quantitative findings shed light on certain characteristics and social identities that may impact the health equity of mobile interventions, more comprehensive research that examines qualitative evidence may complement our findings and solidify our conclusions [74,75]. We recommend that future updates of this review expand the search strategy to consider other nursing and social sciences databases which are abundant with qualitative evidence on the subject matter.

Our findings show the potential mobile interventions have for preventing perinatal depression and psychological stress and managing postpartum depression symptoms once they arise. Future research should focus on examining outcomes of anxiety and utilization of care using more rigorous methods and sufficient sample sizes to gain higher certainty evidence on what works in the field of mobile interventions. Furthermore, our equity findings highlight multiple social characteristics of influence upon the effectiveness of mobile interventions; ethnicity, age, education, and being a first-time mother (primiparous). These findings provide future investigators with the opportunity to focus their research on exploring the impact of such characteristics using an intersectionality lens before implementing mobile interventions in different contexts and settings.

## Conclusion

As the COVID-19 pandemic transitions mental health care delivery into a virtual reality, a knowledge base is needed to inform key knowledge users on what works among pregnant and postpartum women. Our review highlights the effectiveness of mobile interventions and directs future research towards certain characteristics and social identities that require further investigation, such as ethnicity, age, education, and being a first-time mother.

## Supporting information

**S1 File. PRISMA reporting checklist.**
(DOCX)

**S2 File. PRISMA-E reporting checklist.**
(DOCX)

**S3 File. Knowledge translation plan.**
(DOCX)

**S4 File. Search strategy and grey literature outputs.**
(DOCX)

**S5 File. Standardized data extraction form.**
(DOCX)

**S6 File. Critical appraisal visuals.**
(DOCX)

**S7 File. GRADE Evidence profiles.**
(DOCX)

**S8 File. Compartmentalized (Outcome x PROGRESS+) table of equity results.**
(DOCX)

## Acknowledgments

We would like to extend our appreciation to Dr. Vivian Welch and Dr. Melissa Brouwers for their revisions of this work.

## Author Contributions

**Conceptualization:** Ammar Saad, Olivia Magwood, Qasem Alkhateeb, Azaad Kassam, Kevin Pottie.

**Data curation:** Ammar Saad, Olivia Magwood, Qasem Alkhateeb, Syeda Shanza Hashmi, Kevin Pottie.

**Formal analysis:** Ammar Saad, Olivia Magwood, Tim Aubry, Qasem Alkhateeb, Syeda Shanza Hashmi, Julie Hakim, Leanne Ford, Azaad Kassam, Peter Tugwell, Kevin Pottie.

**Funding acquisition:** Ammar Saad, Kevin Pottie.

**Investigation:** Ammar Saad, Tim Aubry, Julie Hakim, Leanne Ford, Azaad Kassam, Peter Tugwell, Kevin Pottie.

**Methodology:** Ammar Saad, Olivia Magwood, Tim Aubry, Julie Hakim, Leanne Ford, Azaad Kassam, Peter Tugwell, Kevin Pottie.

**Project administration:** Ammar Saad.

**Software:** Ammar Saad.

**Supervision:** Peter Tugwell, Kevin Pottie.

**Visualization:** Ammar Saad.

**Writing – original draft:** Ammar Saad, Olivia Magwood, Tim Aubry, Qasem Alkhateeb, Syeda Shanza Hashmi, Azaad Kassam, Kevin Pottie.

**Writing – review & editing:** Ammar Saad, Olivia Magwood, Tim Aubry, Qasem Alkhateeb, Syeda Shanza Hashmi, Julie Hakim, Leanne Ford, Azaad Kassam, Peter Tugwell, Kevin Pottie.

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
