## [Decision Letter · Decision Letter 0]

2 Sep 2021

PONE-D-21-24609

Mobile interventions targeting common mental disorders among pregnant and postpartum women: An equity-focused systematic review

PLOS ONE

Dear Dr. Kevin Pottie,

Thank you for submitting your manuscript to PLOS ONE. After careful consideration, we feel that it has merit but does not fully meet PLOS ONE’s publication criteria as it currently stands. Therefore, we invite you to submit a revised version of the manuscript that addresses the points raised during the review process.

We look forward to receiving your revised manuscript.

Kind regards,

Eugene Demidenko, Ph.D.

Academic Editor

PLOS ONE

Journal Requirements:

Additional Editor Comments:

The study reports on the meta-analysis of mobile interventions aimed to improve mental health of pregnant and postpartum women. I’m overly positive on the work done, accomplishments and findings, but at the same time, I’m frustrated at the lack of transparency of how the summary statistics have been obtained. The paper will become reader-friendly, much stronger and results more convincing upon expanding the quantitative aspect of the paper. I should remind the authors that scientific work requires complete transparency regarding derivation of the results so that they could be reproduced.

Specifically,

1. Unfortunately, the authors mostly report on the qualitative findings but make little effort in explaining how the major quantitative summary statistics such as OR, absolute risk reduction (ARR; why RD?), mean difference (MD) and have been derived.

2. Table 2 misses critical information on quantitative study-specific statistics, such as MD, RD or OR along with their CI, sample size (n) along with statistical method used (logistic regression, contingency table, t-test). When reporting the summary statistics (meta-analysis) from #1 the authors should indicate what data from individual studies have been used.

3. Why Figs 4 and 5 involves only two studies per figure? Why not all 18 studies? What IV, random means?

4. I’m curious why our paper on the mobile intervention “Measuring outcomes of digital technology-assisted nursing postpartum: A randomized controlled trial” by Deborah E McCarter, Eugene Demidenko, Mark T Hegel, PMID: 29772609 PMCID: PMC6240405 DOI: 10.1111/jan.13716 was not a part of the study.

I believe that the authors can address my comments without major revision. I’m looking forward to see the revised version.

Reviewers' comments:

Reviewer's Responses to Questions

**Comments to the Author**

1. Is the manuscript technically sound, and do the data support the conclusions?

Reviewer #1: Yes

2. Has the statistical analysis been performed appropriately and rigorously? 

Reviewer #1: I Don't Know

3. Have the authors made all data underlying the findings in their manuscript fully available?

Reviewer #1: Yes

4. Is the manuscript presented in an intelligible fashion and written in standard English?

Reviewer #1: Yes

5. Review Comments to the Author

Reviewer #1: I can’t comment on the high-level statistics, but as for how it was conceived and reviewed, it is quite thorough. It is important to have searched the nursing database CINAHL, although ultimately only one nursing journal was referenced, and while this may be due to the rigor of the studies, nursing journals are very focused on the clinical and qualitative results of interventions, and are often the ones best qualified to provide the intervention, so more nursing focus could have strengthened this.

The equity lens is excellent, and a great model for others to follow, and adds a significant part to the rigor and value of this research. Similarly, the chart with the risk of bias is quite valuable, but one needs to also consider how this kind of intervention is measured, and what results might be missed by prioritizing randomized controlled trials, which don't lend themselves as well to measuring outcomes of interventions designed to improve mood and mental health. Measuring mental health is limited by the measurement instruments used, and thus, the need for qualitative data, as mentioned in the discussion, is paramount. All in all, I think it was a great paper.

6. PLOS authors have the option to publish the peer review history of their article (what does this mean?). If published, this will include your full peer review and any attached files.

Reviewer #1: **Yes: **Deborah McCarter, PhD, RN

---

## [Author Response · Author response to Decision Letter 0]

16 Oct 2021

Dear Dr. Eugene Demidenko,

We would like to thank you and the reviewers for your careful consideration of our manuscript. The fields of maternal mental health and virtual care are ever-evolving and we envision this project as the first equity-focused intersection between these two fields. The findings that we highlight carry the potential to improve pregnant and postpartum women’s access to timely and appropriate mental health care, and we value the positive comments we have received about the impact of our findings.

Our team has carefully reviewed your revisions and addressed your comments. In summary, we have expanded our methods section to better explain our statistical analysis plan, providing readers with more information on what data was obtained from studies, and how it was analyzed and reported to facilitate reproducibility of our results and ensure transparency of our reporting. We have also reviewed our reference list and ensured that our manuscript meets your journal’s style and submission requirements. All changes are accompanied by a response and marked with the page and line numbers in which changes were made (Please see Response to Reviewers file). 

Should you have any further questions or require further clarifications, please do not hesitate to contact our team.

Kevin Pottie and Ammar Saad on behalf of authors

Response to editor/ reviewer comments:

Journal Requirements:

We have ensured that our manuscript meets PLOS ONE’s style requirements and we named our files in accordance with your submission requirements. 

We have reviewed our reference list to ensure all cited records were correct and no cited records were retracted. The following changes have been made to our reference list:

We have added reference [41] to our list as part of expanding our methods section [Page 22; Line 484]

We have fixed the following 5 references to ensure their completeness and easy accessibility:

Reference [36]: we have added a link to the guidance document [Page 22; Line 474-475]

Reference [39]: We have added a link to the guidance document [Page 22; Line 480-481]

Reference [42]: We have added book chapter page numbers and fixed referencing style [Page 22; Line 485-487]

Reference [53]: We have added a link to the full text manuscript [Page 23; Line 517-518]

Reference [67]: We have added publisher name and year [Page 24; Line 550]

Additional Editor Comments:

The study reports on the meta-analysis of mobile interventions aimed to improve mental health of pregnant and postpartum women. I’m overly positive on the work done, accomplishments and findings, but at the same time, I’m frustrated at the lack of transparency of how the summary statistics have been obtained. The paper will become reader-friendly, much stronger and results more convincing upon expanding the quantitative aspect of the paper. I should remind the authors that scientific work requires complete transparency regarding derivation of the results so that they could be reproduced.

Thank you for sharing our positivity around the findings of this novel equity-focused systematic review, and for highlighting the shortcoming of describing our quantitative data analysis plan. We have now addressed this shortcoming by adding sufficient details to our methods, clearly explaining our statistical analysis plan, including what data was extracted from included studies, and how such data was analyzed and presented [Page 5-6; Line 126-139].

Specifically,

1. Unfortunately, the authors mostly report on the qualitative findings but make little effort in explaining how the major quantitative summary statistics such as OR, absolute risk reduction (ARR; why RD?), mean difference (MD) and have been derived.

We have added to our methods section to explain how quantitative summary statistics were derived/ calculated for both continuous and categorical outcomes [Page 5-6; Line 126-139].

2. Table 2 misses critical information on quantitative study-specific statistics, such as MD, RD or OR along with their CI, sample size (n) along with statistical method used (logistic regression, contingency table, t-test). When reporting the summary statistics (meta-analysis) from #1 the authors should indicate what data from individual studies have been used.

We have created a continuation of Table 2 to present information about study-specific statistics, such as the sample size, study-level statistical method used, study results retrieved from each included study, and effect estimates calculated for the purpose of our data synthesis [Page 12-13; Line 195].

Furthermore, we have added to our methods to explain what data was used from individual studies that contributed to our Meta-Analyses (i.e., mean differences at follow-up) [Page 6; Line 147-149]. The information provided in the continuation of Table 2 should provide readers of what values were used to calculate these mean differences from each included study [Page 12-13; Line 195].

3. Why Figs 4 and 5 involve only two studies per figure? Why not all 18 studies? What IV, random means?

While we recognize that meta-analyses with a larger number of studies provide more precise pooled results, ensuring that these meta-analyses are built on sound methods is paramount. Unfortunately, the clinical heterogeneity between studies prevented pooling data from more than the two studies in each of Fig 4 and 5. Clinical heterogeneity arose due to the different pregnancy stages that women were in when the intervention was delivered, different intervention designs, and different outcome measurement tools. We have added this explanation to our results to provide readers with more information about lack of studies in the two meta-analyses [Page13; Line 218-220 and Page 15; Line 255-258].

“IV, Random” is an automatic label created by our analysis software (RevMan 5.4) to indicate the statistical method used in pooling results (i.e., Inverse Variance) and analysis model (i.e., Random effects model). Thank you for pointing out the need to clarify that. We have added to the legends of Figs 5 and 6 to explain that label [Page 14; Line 243 and Page 16; Line 276].

4. I’m curious why our paper on the mobile intervention “Measuring outcomes of digital technology-assisted nursing postpartum: A randomized controlled trial” by Deborah E McCarter, Eugene Demidenko, Mark T Hegel, PMID: 29772609 PMCID: PMC6240405 DOI: 10.1111/jan.13716 was not a part of the study.

Our records show that your publication titled “Measuring outcomes of digital technology-assisted nursing postpartum: A randomized controlled trial” was indeed captured by our search and screened in duplicate by our team. However, Phases 1 and 2 were excluded from our final list of included studies due to “ineligible study design” (i.e., one-arm studies with no control group). While the third phase of the study is an open-label three-arm parallel RCT, the publication reports that follow-up and/ or analysis is still ongoing and only shares baseline demographics. Therefore, we have labelled your study as “ongoing”.

I believe that the authors can address my comments without major revision. I’m looking forward to see the revised version.

We thank you again for your comments and hope that we have addressed them to your standards. If you have any further comments or require any further clarifications, please do not hesitate to contact us.

Reviewers' comments:

Reviewer #1: I can’t comment on the high-level statistics, but as for how it was conceived and reviewed, it is quite thorough. It is important to have searched the nursing database CINAHL, although ultimately only one nursing journal was referenced, and while this may be due to the rigor of the studies, nursing journals are very focused on the clinical and qualitative results of interventions, and are often the ones best qualified to provide the intervention, so more nursing focus could have strengthened this.

We thank you, Dr. McCarter, for reviewing and highly appraising our work. We share your perspective on the importance of examining mobile interventions using an interdisciplinary lens that considers the field of nursing sciences, among others, in curating evidence on the subject matter. Our iterative process of developing the search strategy with the health sciences librarian at the University of Ottawa as well as experts in knowledge syntheses identified CINAHL as the database of choice for its inclusivity of nursing publications and records that meet our study design inclusion criteria following the Cochrane Effective Practice and Organization of Care guidelines (i.e., experiments with a controlled arm). You mentioned that the methodological rigour of our inclusion criteria may have hindered including more studies from the field of nursing and we agree, recognizing that the field of nursing sciences may include further publications of longitudinal one-arm studies as well as qualitative studies on the subject matter. We have added to our discussion [Page 19; Line 359-361] to recommend adding more nursing-specific databases and study designs to future updates of this review. 

The equity lens is excellent, and a great model for others to follow, and adds a significant part to the rigor and value of this research. Similarly, the chart with the risk of bias is quite valuable, but one needs to also consider how this kind of intervention is measured, and what results might be missed by prioritizing randomized controlled trials, which don't lend themselves as well to measuring outcomes of interventions designed to improve mood and mental health. Measuring mental health is limited by the measurement instruments used, and thus, the need for qualitative data, as mentioned in the discussion, is paramount. All in all, I think it was a great paper.

Our vision for this review is to serve as a blueprint for future work that considers and addresses equity around maternal mental health interventions, specifically ones that utilize virtual or digital means of care delivery. We thank you for highlighting the significance of the equity lens. In regards to your comment about study designs and outcome measurements, one of the lengthy debates we have had among our team and with our equity experts is the balance between including evidence of higher certainty from studies with rigorous methodology (e.g., Randomized and non-randomized controlled trials) versus enriching our findings with broader but less certain evidence from other study designs (e.g., Observational and qualitative studies). While there is no right or wrong approach to either of these two arguments, we elected to consider this review as a first step, in which we present our knowledge users, such as healthcare providers, patients, and policy makers with higher certainty evidence that shows the effectiveness and equity potential of mobile interventions to raise awareness and draw attention to this field, while highlighting the importance of including broader evidence in future updates of this work. To highlight the need for future inclusion of other study design, we have highlighted, in our discussion, the need to expand the eligibility criteria in future updates of this review [Page 19; Line 353-357].

6. PLOS authors have the option to publish the peer review history of their article (what does this mean?). If published, this will include your full peer review and any attached files. Do you want your identity to be public for this peer review? For information about this choice, including consent withdrawal, please see our Privacy Policy.

Reviewer #1: Yes: Deborah McCarter, PhD, RN

We have uploaded our figures to the Preflight Analysis and Conversion Engine (PACE) digital diagnostic tool and they have been adjusted according to your journal’s requirements. We have uploaded the adjusted figures to our resubmission.

---

## [Editor Report · Decision Letter 1]

20 Oct 2021

Mobile interventions targeting common mental disorders among pregnant and postpartum women: An equity-focused systematic review

PONE-D-21-24609R1

Dear Dr. Pottie,

We’re pleased to inform you that your manuscript has been judged scientifically suitable for publication and will be formally accepted for publication once it meets all outstanding technical requirements.

Kind regards,

Eugene Demidenko, Ph.D.

Academic Editor

PLOS ONE

Additional Editor Comments (optional):

The authors addressed the comments raised by the Editor, who acted as a reviewer, and another reviewer. The paper is in publishable form. Just an advice to make the paper reader-friendly: in the two tables where you indicate 'statistical significance' please remind that this means that p-value < 0.05 (correct?).

---

## [Editor Report · Acceptance letter]

22 Oct 2021

PONE-D-21-24609R1 

Mobile interventions targeting common mental disorders among pregnant and postpartum women: An equity-focused systematic review 

Dear Dr. Pottie:

I'm pleased to inform you that your manuscript has been deemed suitable for publication in PLOS ONE. Congratulations! Your manuscript is now with our production department. 

Kind regards, 

on behalf of

Dr. Eugene Demidenko 

Academic Editor

PLOS ONE